# Primary TSC2^-/meth^ Cells Induce Follicular Neogenesis in an Innovative TSC Mouse Model

**DOI:** 10.3390/ijms23179713

**Published:** 2022-08-26

**Authors:** Clara Bernardelli, Eloisa Chiaramonte, Silvia Ancona, Silvia M. Sirchia, Amilcare Cerri, Elena Lesma

**Affiliations:** 1Laboratory of Pharmacology, Department of Health Sciences, Università degli Studi di Milano, 20142 Milan, Italy; 2Medical Genetics, Department of Health Sciences, Università degli Studi di Milano, 20142 Milan, Italy; 3Dermatology Unit, Department of Health Sciences, Università degli Studi di Milano, 20142 Milan, Italy

**Keywords:** tuberous sclerosis complex, primary cells, follicular neogenesis, cellular migration, mTOR, 5-azacytidine

## Abstract

Cutaneous lesions are one of the hallmarks of tuberous sclerosis complex (TSC), a genetic disease in which mTOR is hyperactivated due to the lack of hamartin or tuberin. To date, novel pharmacological treatments for TSC cutaneous lesions that are benign but still have an impact on a patient’s life are needed, because neither surgery nor rapamycin administration prevents their recurrence. Here, we demonstrated that primary TSC2^-/meth^ cells that do not express tuberin for an epigenetic event caused cutaneous lesions and follicular neogenesis when they were subcutaneously injected in nude mice. Tuberin-null cells localized in the hair bulbs and alongside mature hairs, where high phosphorylation of S6 and Erk indicated mTOR hyperactivation. Interestingly, 5-azacytidine treatment reduced hair follicles, indicating that chromatin remodeling agents might be effective on TSC lesions in which cells lack tuberin for an epigenetic event. Moreover, we demonstrated that the primary TSC2^-/meth^ cells had metastatic capability: when subcutaneously injected, they reached the bloodstream and lymphatics and invaded the lungs, causing the enlargement of the alveolar walls. The capability of TSC2^-/meth^ cells to survive and migrate in vivo makes our mouse model ideal to follow the progression of the disease and test potential pharmacological treatments in a time-dependent manner.

## 1. Introduction

Tuberous sclerosis complex (TSC) is a rare multisystemic monogenic disease caused by a germline mutation in either *TSC1* or *TSC2* genes, followed by a second genetic/epigenetic hit inactivating the wild-type allele. *TSC1* and *TSC2* encode hamartin and tuberin, respectively [1]. These proteins form a complex that negatively regulates the mammalian target of rapamycin complex 1 (mTORC1), a central regulator of cell growth and proliferation. mTOR integrates cellular nutrient supply and stress status, inducing appropriate cellular responses; its deregulation occurs in many human diseases, including cancer [2]. The lack of hamartin or tuberin leads to the constitutive activation of mTORC1 in TSC-associated lesions, which include cutaneous lesions, renal angiomyolipoma (AML), intraventricular subependymal giant cell astrocytoma, cardiac rhabdomyomas, and pulmonary lesions, whose main manifestation in women is lymphangioleyomiomatosis (LAM), a rare progressive cystic lung disease that can occur in a sporadic form or associated to TSC [3]. Cutaneous lesions, including facial angiofibromas, fibrous plaques, hypomelanotic macules, and periungual fibromas, are a hallmark to diagnose TSC, as they occur in almost all TSC patients [4]. These lesions are hamartomatous, localized anywhere on the skin, but most frequently on the face, trunk, buttocks, or upper thighs, and show a disorganized set of fibroblast-like cells, fibrous tissue, and blood vessels. Cutaneous lesions are benign and not life-threatening, but they may worsen the quality of life of the patients. In addition to an extensive number of observational reports, the underlined development mechanism of TSC dermatological lesions is not completely understood, likely also because of their cellular heterogeneity and genetic complexity [5]. The treatment of TSC-associated skin lesions includes surgery and oral or topical administration of mTOR inhibitors, even if they do not prevent their recurrence [6]. For this reason, a model of TSC cutaneous lesions reproducing the complexity of the genetic and epigenetic events that drive their onset is needed to better understand the mechanisms underlying lesion formation and test the long-term follow-up of innovative pharmacological treatments.

Here, we developed a mouse model for skin manifestations by subcutaneous injection of TSC2^-/meth^ cells in nude mice. The cells were primary tuberin-null cells previously isolated from a renal AML of a TSC patient [7]. TSC2^-/meth^ cells display the ability to survive in vivo and, from skin, to systemically migrate to tissues. Notably, in our human model, primary TSC2^-/meth^ cells caused skin alterations and induced follicular neogenesis. As TSC2^-/meth^ cells display an epigenetic second hit due to the hypermethylation of the *TSC2* promoter [7], this study shows, for the first-time, preliminary data of the in vivo use of the chromatin remodeling agent 5-azacytidine for the treatment of a TSC-related manifestation. To conclude, our mouse model reproduces the genetic and epigenetic complexity underlying the onset and progression of skin-TSC-related manifestations, making primary cell injection a suitable tool for the study of cutaneous TSC lesions.

## 2. Results

### 2.1. TSC2^-/meth^ Cells Induce Skin Alterations and Follicular Neogenesis in Nude Mice

PKH26-labeled TSC2^-/meth^ cells were subcutaneously injected on the back or flank of female athymic nude-fox1nu mice that were 12–14 weeks old. Starting from 30 days after cell injection, a skin thickening was developed on the neck and around the tail, with an enrichment of PKH26-labeled TSC2^-/meth^ cells (Figure 1a) of approximately 6.889 ± 1.809 cells/mm^2^ or 654.2 ± 171.8 cells/mm^3^ (Figure 1b). These thickened areas were not necessarily near the injection site, where the average number of PKH26-labeled cells was 0.759 ± 0.265 cells/mm^2^ or 72.09 ± 25.23 cells/mm^3^ (Figure 1b), and the cells did not have the appearance of a solid tumor mass; indeed, TSC2^-/meth^ cells were distributed along all the skin section. Furthermore, PKH26-labeled TSC2^-/meth^ cells were detected in skin regions that did not show any alterations at the macroscopic level, even far from the injection site or the thickened skin areas (Figure 1a), suggesting a migratory capability of tuberin-null cells in the dermal tissue. In particular, the average number of TSC2^-/meth^ cells in nonaltered skin was 1.253 ± 0.627 cells/mm^2^ or 119.0 ± 59.56 cells/mm^3^. Macroscopically, the aspect of the thickened skin resembled the shagreen patches lesions observed in TSC patients. Furthermore, the thickened areas were characterized by massive hair growth, which is atypical in nude mice and was not observed in the controls (Figure 1c).

The results of immunohistochemistry analysis highlighted the formation of hair bulbs in mice injected with tuberin-null cells (Figure 2a). TSC2^-/meth^ cells localized in the hair bulbs and alongside the mature hair, as demonstrated by their positivity to the human-specific HMB45 antibody, which is considered a marker of TSC cells [8] (Figure 2b).

Hamartin–tuberin complex regulates mTORC activity, which phosphorylates S6 kinase (S6K) 1, ultimately promoting cell growth and proliferation [9]. Following the injection of TSC2^-/meth^ cells, we observed high S6 phosphorylation in the thickened skin, near hair bulbs, and alongside mature hair, indicating high mTOR activity (Figure 3a). Moreover, high levels of phosphorylated extracellular-signal-regulated kinase (Erk), which is considered a functional marker of TSC cells in the hyperactivation of S6, were detected in the hair bulbs and alongside mature hair and, in addition, in the subcutaneous layer, as observed for phospho-S6 (Figure 3b).

More specifically, we estimated that 55.61% (±SEM 0.589) of the cells in the skin section of mice injected with TSC2^-/meth^ cells were positive for phospho-S6 compared with the control mice (21.51% ± 0.656), which is a significant difference (Student’s *t*-test, *p*-value < 0.0001). Similarly, the percentage of phospho-Erk positive cells was 34.01% (±2.654) in mice that received TSC2^-/meth^ cells versus 18.42% ± 2.385% in the controls (Student’s *t*-test, *p* value < 0.05) (Appendix A). TSC2^-/meth^ cells do not express tuberin for an epigenetic event that was demonstrated by the transcriptional reactivation of *TSC2* after the treatment with chromatin-remodeling agent 5-azacytidine, which promotes tuberin expression in TSC2^-/meth^ cells [7]. For this reason, 60 days after cell injection, the mice were treated once a day for 4 days with 5-azacytidine (2 mg/kg/day, intraperitoneally administered). The pharmacological treatment caused a reduction in hair bulbs and a decrease in phospho-S6 (33.80% ± 1.776%, *p* < 0.0001 vs. TSC2^-/meth^-cells-injected mice) and phospho-ERK (24.65% ± 1.265, *p* < 0.05 vs. TSC2^-/meth^ cells injected mice)-positive cells (Figure 4, Appendix A).

### 2.2. TSC2^-/meth^ Cells Subcutaneously Injected Can Migrate In Vivo

To evaluate the systemic migratory capability of TSC2^-/meth^ cells following subcutaneous injection, we took advantage of the fact that they were isolated from a male patient to detect, by PCR analysis, the SRY sequences of the human Y chromosome in the whole bloodstream of the female injected mice. The expression of the SYR sequences indicated the presence of male human cells in the female mice’s blood (Figure 5a). Interestingly, after 60 days from the injection, we also confirmed the presence of TSC2^-/meth^ cells in the lymph nodes through the detection of the red fluorescent signal of PKH26-labeled cells (Figure 5b). These results demonstrated that, as in humans, TSC cells have migratory capability and, furthermore, exploit the lymphatic system and blood stream to invade the body.

As LAM is the main pulmonary manifestation in female TSC patients [3] and because TSC2^-/meth^ cells can efficiently migrate in the dermal tissue (Figure 1a), we investigated the capability of TSC2^-/meth^ to reach the lung parenchyma. Red fluorescent PKH26-labeled TSC2^-/meth^ cells were detected in the lung alveoli once subcutaneously injected (Figure 6a). As in the previous LAM-developed models, in which tuberin-null cells were inhaled [10], TSC2^-/meth^ cells induced degeneration of the lung parenchyma by disrupting the alveolar walls and enlarging the alveolar spaces, however without creating solid tumoral mass (Figure 6b). The administration of 5-azacytidine reverted the lung parenchyma alterations and induced a massive thickening of the alveolar walls (Figure 6c), a known side effect of this drug [11]. The morphological alterations in lung parenchyma were quantitatively evaluated by the density of air-exchanging parenchyma [*A_A_* (ae/lu)] as previously described [12] (Figure 6d).

## 3. Discussion

Skin lesions are one of the most common manifestations of TSC, a rare autosomal-dominant disease characterized by hamartomas in different organs including the brain and kidneys, in addition to skin [4]. TSC cutaneous lesions are not life-threatening, but they strongly affect the quality of life of patients and, applying the state of the art methods, the approved treatments (mainly surgery and topical administration of mTOR inhibitors) do not prevent their recurrence [6].

We developed a mouse model of TSC skin lesions by the subcutaneous injection of human primary TSC2^-/meth^ cells, isolated from an AML of a TSC patient. Different from other primary TSC cells, TSC2^-/meth^ cells display an epigenetic second hit due to the methylation of the CpG island of the *TSC2* promoter, leading to the inhibition of tuberin expression [6]. Sixty days after the subcutaneous injection of PKH26-labeled TSC2^-/meth^ cells in the female athymic nude-fox1nu mice, we observed a thickening of the skin in different areas of the mouse back that, in some cases, showed massive hair growth. These thickened areas resembled the so-called shagreen patches observed in TSC patients. Moreover, TSC2^-/meth^ cells migrated in the mouse skin from the injection site without any cellular accumulation as observed in other published models in which the subcutaneous injection of tuberin-null cells induced the formation of a solid mass [13]. Skin thickening was already reported in a TSC mouse model obtained by biallelic deletion of the *Tsc1* gene in fibroblasts [14]; yet, until now, no one has reproduced the TSC skin lesions by subcutaneous injection of human primary TSC cells. Remarkably, in our model, we also observed hair growth, consistent with the evidence reported in a TSC model of skin lesions obtained through the xenograft of a composite made of tuberin-null fibroblast-like cells and normal human keratinocytes [13]. Of note, different from this model, the origin of TSC2^-/meth^ cells, the AML, is not expected to induce hair follicular neogenesis, thus supporting the finding that the absence of tuberin caused the development of the skin manifestation. The TSC2^-/meth^ cells’ localization (inside hair bulbs, alongside mature hair, and in the thickened skin) supports the hypothesis that cutaneous hamartomas express mosaicism for *TSC2* second-hit events due to the heterogeneity of the cells in the lesions [14]. It is possible that only a small part of keratinocytes, or derma fibroblasts, have a mutation in the *TSC2* second allele, being capable of inducing an uncontrolled proliferation and abnormal behavior in the surrounding cells of the lesion [15].

The constitutive activation of mTORC1 due to the lack of tuberin expression leads to the phosphorylation of S6K1, ultimately promoting cell growth and proliferation [16]. In mice injected with TSC2^-/meth^ cells, high S6 phosphorylation was observed in the altered areas, including thickened skin near hair bulbs and alongside mature hair. This result is consistent with the hypothesis that mTOR might drive hair follicle tumorigenesis [17]. Moreover, the phosphorylation of Erk that, together with S6 activation, might be considered a functional marker of TSC cells [18], was high in the skin of mice that received tuberin-null cells. The distribution of cells expressing phospho-S6 and phospho-Erk in mice injected with TSC2^-/meth^ cells resembles what we observed in fibromas isolated from TSC patients, where these proteins are mainly expressed in the subcutaneous, as previously reported for human TSC hamartomas [19]. Both patients’ fibrous plaques and angiofibroma show hair follicles that are enlarged, elongated, and increased in number [13]. Of note, the phospho-S6- and phospho-Erk-positive cells were distributed in the hair follicles and surrounding area, not necessarily with the same TSC2^-/meth^ cell localization, suggesting that tuberin-null cells might also induce mTOR hyperactivation in tuberin-expressing cells. This mechanism, however, still needs to be investigated and is consistent with the hypothesis that *Tsc1*- or *Tsc2*-null cells mediate the acquisition of the disease phenotypes by cells with normal genome through exosome release [20].

The epigenetic silencing of TSC2^-/meth^ cells that leads to the inhibition of tuberin transcription can be reactivated by the chromatin-remodeling agents, such as 5-azacytidine, to restore tuberin expression, indicating that those molecules might act as potential therapeutic agents [7]. To study the in vivo effects of chromatin-remodeling agents, we treated mice with 5-azacytidine 60 days after tuberin-null cell injection. At the cutaneous level, we observed a reversion of both skin thickening and hair growth. Phospho-S6- and phospho-Erk-positive cells in skin sections were also decreased. It was demonstrated that mTOR hyperactivation is involved in malignant hair follicle tumorigenesis [17]; so, in our model, it is possible that the transcriptional reactivation of *TSC2* impairs the capability of TSC2^-/meth^ cells that preferentially localize in the hair bulbs to induce hair follicle formation. TSC2^-/meth^ cells demonstrated an intrinsic metastatic capability. They were detected in the blood and lymph nodes of mice 60 days after their injection, indicating that tuberin-null cells migrate through the bloodstream and lymphatic system, as we previously demonstrated in an innovative mouse model of LAM [10]. The preferential infiltration of TSC2^-/meth^ cells in the lungs is a finding in accordance with that in our previous study in which a LAM mouse model was developed by the endonasal administration of TSC2^-/-^ cells and LAM/TSC cells [10,12]. However, in those models, the administration route was specifically chosen to allow cells to reach the lungs. The presence of TSC2^-/meth^ cells in the lungs following subcutaneous injection suggests a tropism of the tuberin-null cells for the lung. TSC2^-/meth^ cells infiltrated the lung, causing a thickening of the parenchyma, and a disruption of the alveolar walls, leading to the enlargement of the alveolar spaces. This condition resembles what is observed in the lungs of TSC patients, in particular when tuberous sclerosis is associated to LAM [3], where the involvement of the lungs is bilateral without evidence of a primary tumor site [21]. As further support of the reproducibility of our model, we observed that the subcutaneous injection of primary LAM/TSC cells induced cutaneous lesions and follicular neogenesis similarly to what was observed for TSC2^-/meth^ cells. With the treatment with 5-azacytidine, the enlargement of the air spaces reverted toward normal but caused an enhanced thickness of the parenchyma, which may be a side effect. In a murine model of orthotopic lung cancer, the intratracheal administration of 5-azacytidine caused an initial state of inflammation with pneumocyte hypertrophy, which reverted 14 days after the therapy suspension while maintaining an effective reversion of the hypermethylation status of the genes of lung cancer cell lines [11]. In conclusion, our study demonstrated for the first time that primary tuberin-null cells isolated from patients induce follicular neogenesis that might be driven by mTOR hyperactivation. When *TSC2* epigenetic silencing as a second hit inactivates the wild-type allele as in the case of TSC2^-/meth^ cells, chromatin remodeling agents, such as 5-azacytidine, can revert specific pathological manifestations. For this reason, our model will shed light on new potential pharmacological approaches for TSC, such as targeting the epigenetic silencing of the *TSC2* promoter or interfering with the capability of TSC cells to communicate with their microenvironment.

## 4. Materials and Methods

### 4.1. Cell Culture

TSC2^-/meth^ cells were isolated, characterized, and grown as previously described [7]. These cells were obtained from an AML of a TSC male patient (36 years old) who had given his informed consent according to the Declaration of Helsinki. The study was approved by the Institutional Review Board of Milan’s San Paolo Hospital [7]. TSC2^-/meth^ cells were grown in a Type II medium, containing a mixture of DMEM/Ham F12 (50:50 ratio) (Euroclone, Paignton, U.K) supplemented with 15% fetal bovine serum (Euroclone), 1% glutamine (Euroclone), 2% penicillin/streptomycin (Euroclone), 200 nM hydrocortisone (Sigma-Aldrich, St. Louis, MO, USA), 10 ng/mL epidermal growth factor (EGF, Sigma-Aldrich), and 1.6 μM ferrous sulfate (Sigma-Aldrich).

In all the experiments, TSC2^-/meth^ cells were cultured for 3 up to 5 passages (meaning for about 20 days after the thawing), as we previously described [22]. This precaution allowed us to maintain and standardize the primary features of these cells.

### 4.2. Animal Experiments and Pharmacological Treatments

The immunodeficient female nude mice (athymic nude-fox1nu) were obtained from Harlan Laboratories (Udine, Italy). All experimental procedures were performed in accordance with the Italian Guidelines for Laboratory Animals, which conform to the European Committees Directive (86/609/EEC), and the study protocol was previously approved by the Ministero della Sanità (2/2011 Protocol). The mice were housed in individual plastic cages under controlled conditions (temperature: 24 °C, humidity: 65%, light/dark cycle: 12/12 h) with food and water ad libitum.

TSC2^-/meth^ cells were labeled with PKH26-GL using Red Fluorescent Cell Linker kits (Sigma-Aldrich) according to the manufacturer’s instructions [10]. Briefly, 3 × 10^6^ cells were labeled with PKH26 dye (at a final concentration of 4 × 10^−6^ M) and resuspended in 200 μL of physiological solution (0.9% NaCl) before being administered to the mice by subcutaneous injection on the back. Sixty days after the cells injection the mice were sacrificed; and skin, lymph nodes and lungs were removed [12]. Sixty days after cell injection, 5-azacytidine (5-Aza) (Sigma-Aldrich) was intraperitoneally administered in 100 μL at the final concentration of 2 mg/Kg/day for 4 days, according to the dosage reported in the literature [23]. The mice were followed for another 4 days before being sacrificed. All specimens were fixed in 4% paraformaldehyde at 4 °C overnight and embedded in paraffin.

### 4.3. Immunohistochemistry and Fluorescence Analysis

Paraffin-embedded tissues were sectioned (thickness of 5 μm), deparaffinized, and rehydrated in xylene and decreasing concentrations of ethanol to distilled water (100%, 95%, 90%, 80%, and 70%). Immunohistochemistry analysis was performed as previously described [24]. Briefly, slides were treated under pressure at 95 °C in citrate buffer, pH 6.0 for 5 min for antigen retrieval, and incubated with 0.3% H_2_O_2_ in methanol for 20 min to quench endogenous peroxidase activity. The block of nonspecific binding sites was achieved by incubation with 3% albumin bovine serum (BSA) (Sigma-Aldrich) in TBS at room temperature for 2 h. The primary antibody against phospho-S6 (Ser235-236) (1:50; Cell Signaling Technology, Beverly, MA, USA), phospho-p44/42 MAPK (Erk1/2) (Thr202/Tyr204) (1:100; Cell Signaling Technology), and HMB45 (1:50; Dako, Glostrup, Danmark) were incubated in 1.5% BSA at 4 °C overnight, followed by incubation with the appropriate secondary antibody for 2 h at room temperature. For negative controls, the primary antibody was omitted. An ABC Ultrasensitive Peroxidase Staining kit (Pierce, Rockford, IL, USA) or ABC-AP Vector Laboratories (Burlingame, CA) was used for signal enhancement. Color development was achieved by peroxidases substrate DAB (3.3-diaminobenzidine) (DAB substrate kit, Pierce) or vector red alkaline phosphatase substrate; and the sections were finally counterstained with Mayer’s hematoxylin, dehydrated, and mounted. The images were acquired with a LEICA DM4000 B bright-field microscope (Leica microsystems, Wetzlar, Germany) under identical conditions of magnification and illumination.

To quantify phospho-S6- and phospho-Erk-positive cells, the percentage of stained cells (brown cells) over the total amount of nuclei was counted at 40× magnification for at least 3 mice in each group (control mice, mice injected with TSC2^-/meth^ cells, and mice injected with TSC2^-/meth^ cells and treated with 5-azacytidine).

To detect PKH-26-positive cells in the skin and lymph nodes sections, nuclei were stained with 2 μg/mL DAPI (Sigma-Aldrich) after antigen retrieval. Images were acquired with a LEICA DM4000 B fluorescence microscope. The number of PKH26-positive TSC2^-/meth^ cells in the skin sections was determined as previously described [10]. Briefly, we counted at least 3 randomly selected fields for each treatment group. The TSC2^-/meth^ cell number is expressed as cells per millimeter squared or as cells per millimeter cubed, according to the following formula: (TSC2^-/meth^ cells/mm^2^) × [1000/(5.53 µm + 5 µm)], where 5.53 µm represents the mean cell diameter, and 5 µm is the section thickness.

### 4.4. PCR Analysis

Genomic DNA was extracted using a QIAamp DNA Mini and Blood Mini Kit (Quiagen, Hilden, Germany), according to the manufacturer’s instructions. PCR reaction was performed on 50 ng of the sample by using 25 pmol/µL of SRY-specific primers: F: 5′-CAGTGTGAAAGCGGAGAAAACAGT-3′; R: 5′-CTTCCGACGAGGTCGATACTTATA-3′. Amplified sequences were detected by electrophoresis on 2% agarose gel.

### 4.5. Stereological Analysis of Hair-Exchanging Parenchyma

The density of air-exchanging parenchyma, *A_A_* (ae/lu), was manually determined, as previously described [12]. Briefly, we overlapped a 2 cm × 2 cm grid on 40X acquired lung images (*n* = 10 images for 3 independent experiments) and counted the number of intersection points falling on the air-exchanging parenchyma (excluding airspace) over the total intersection points in the grid. Airways, vascular structures, and histological mechanical artifacts were eliminated from the analysis.

### 4.6. Statistical Analysis

Statistical analysis was performed with GraphPad Prism 5 (version 5, GraphPad software, Inc.). *p*-values less than 0.05 were considered statistically significant in a one-way ANOVA.

## Figures and Tables

**Figure 1 ijms-23-09713-f001:**
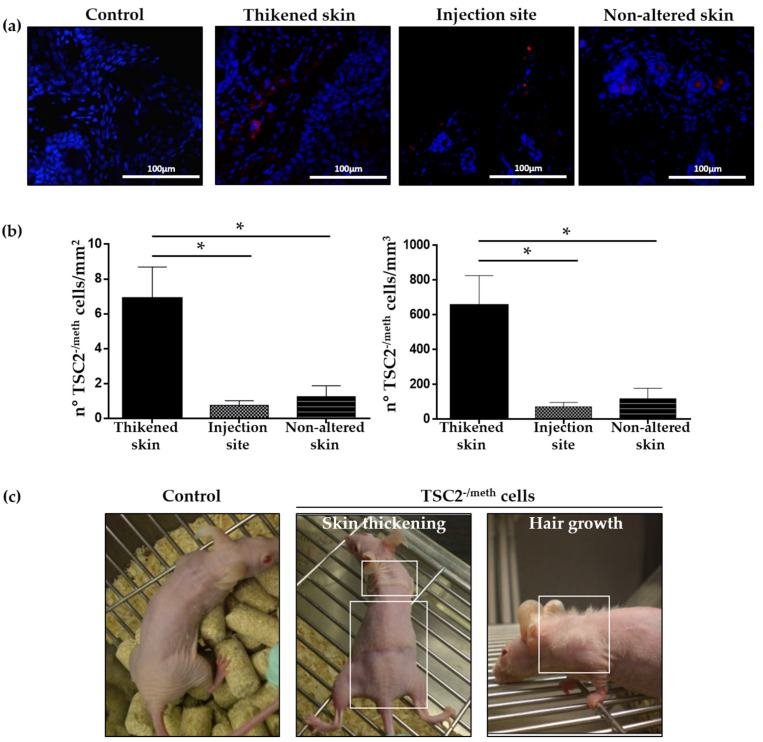
(**a**) Representative images of PKH26-labeled cells (red) detected in the skin of mice 60 days after the injection. Control mice were not injected with TSC2^-/meth^ cells. Nuclei were stained with DAPI. Scale bar: 100 µm (**b**) Quantification of the skin invasion by TSC2^-/meth^ cells in different areas. Data are shown as number of PKH26-labeled cells per millimeter squared (left panel) or as number of PKH26-labeled cells per millimeter cubed (right panel). ANOVA with Tukey’s test. * *p* < 0.05, results are shown as mean ± SEM (**c**) Representative images of skin lesions observed in mice after 60 days from subcutaneous injection of TSC2^-/meth^ cells. White squares highlight thickened areas and hair growth.

**Figure 2 ijms-23-09713-f002:**
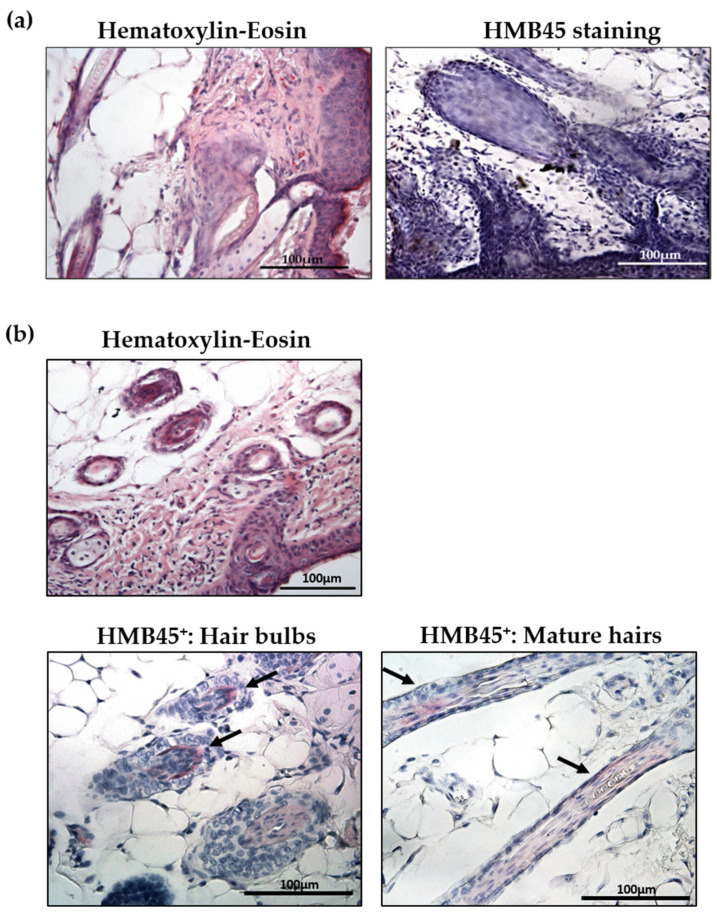
(**a**) Representative hematoxylin–eosin (**left**) and HMB45 immunohistochemistry (**right**) of skin sections from control mice. (**b**) Representative hematoxylin–eosin (upper panel) and HMB45 immunohistochemistry (lower panels) of skin section of mice injected with TSC2^-/meth^ cells. HMB45-positive cells were stained in red (highlighted by arrows). Nuclei were stained with hematoxylin. Scale bar: 100 µm.

**Figure 3 ijms-23-09713-f003:**
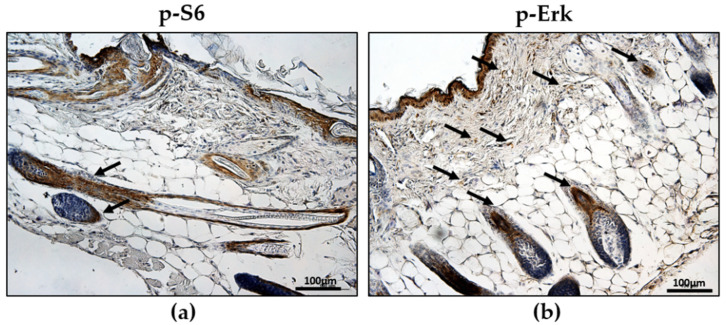
Representative immunohistochemistry for phospho-S6 (**a**) and phospho-Erk (**b**) positivity in the skin of mice injected with tuberin-null cells. Positive cells were stained in brown (highlighted by arrows); nuclei were stained with hematoxylin. Scale bar: 100 µm.

**Figure 4 ijms-23-09713-f004:**
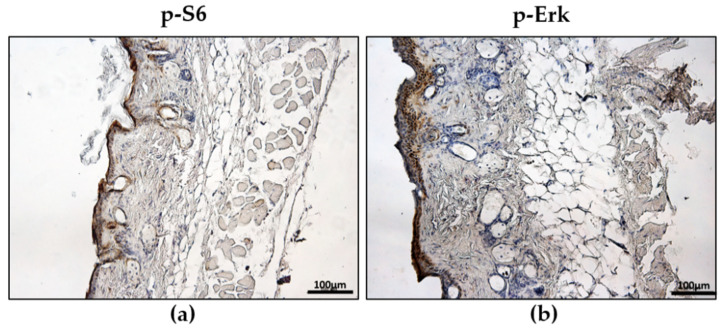
Representative immunohistochemistry for phospho-S6 (**a**) and phospho-Erk (**b**) positivity in the skin of TSC2^-/meth^-cells-injected mice after treatment with 5-azacytidine for 4 days. Nuclei are stained with hematoxylin. Scale bar: 100 µm.

**Figure 5 ijms-23-09713-f005:**
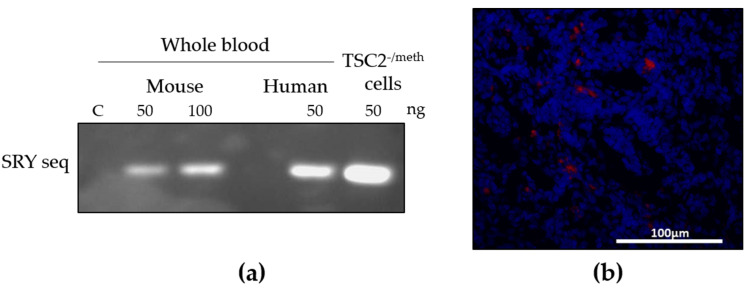
(**a**) Electrophoresis of 50 and 100 ng of PCR amplification product to detect human-specific SRY sequences in the whole blood of a mouse injected with TSC2^-/meth^ cells (Mouse); in the whole blood of a human male subject (Human); and in DNA extracted from TSC2^-/meth^ cells. C: negative control (C). (**b**) Representative images of lymph nodes section, 60 days after TSC2^-/meth^ cells injection. Red fluorescent PKH26-positive cells were detected; nuclei were stained with DAPI. Scale bar: 100 µm.

**Figure 6 ijms-23-09713-f006:**
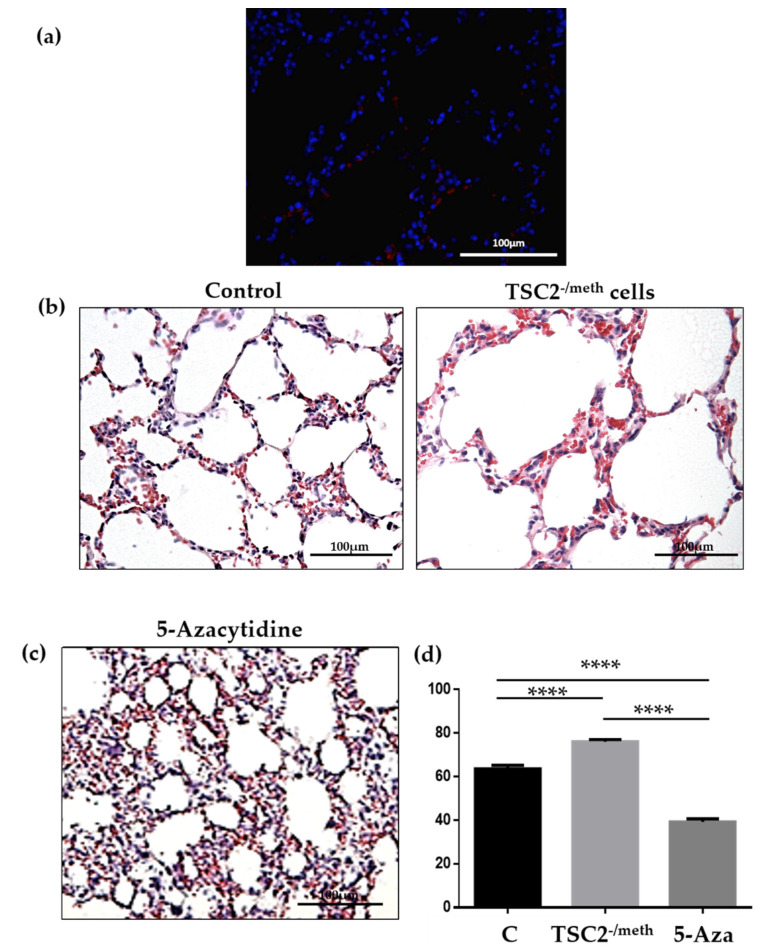
(**a**) Red fluorescent PKH26-positive cells were detected in lung parenchyma 60 days after the injection. Nuclei were stained with DAPI. (**b**) Representative hematoxylin–eosin staining on lung section of control mice (left) and mice injected with TSC2^-/meth^ cells (right). (**c**) Representative hematoxylin–eosin staining on lung section of mice treated with 5-azacytidine. Scale bar: 100 µm. (**d**) Quantification of the fraction of the hair-exchanging parenchyma (*A_A_* (ae/lu)) in *n* = 3 mice per condition, expressed as a percentage (C: control; TSC2^-/meth^: mice injected with tuberin-null cells; 5-Aza: mice injected with tuberin-null cells and treated with 5-azacytidine for 4 days, 60 days after cells injection). ANOVA with Tukey’s test. **** *p* < 0.0001; results are shown as mean ± SEM.

## Data Availability

The data presented in this study are available on request from the corresponding author.

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
