# Peer review of "Primary TSC2-/meth Cells Induce Follicular Neogenesis in an Innovative TSC Mouse Model"

_ijms, 2022, doi:10.3390/ijms23179713_

Round 1
Reviewer 1 Report
I have endorsed this revised version.
Author Response
We thank the Reviewer for the endorsement of the revised version.
Reviewer 2 Report
Tuberous sclerosis is a very interesting field of research. The authors establish a very interesting model of TSC in mice and show the efficacy of 5-azacytidine in case of epigenetical silencing of the tubers gene. The paper is globally well written and well presented. Discussion is consistent with the results. Methods are clearly described. Globally, Bernardelli et al made an excellent job. However, the authors state that their mouse model is ideal to follow the progression of TSC. A short sentence regarding the follicular localization of cutaneous hamartomas and the frequent involvement of the lungs in disseminated forms of the disease is needed for non-experts in the field both in the abstract and in the introduction. Since the two parameters considered in vivo are hair growth and skin thickening, a mention of the clinical relevance of such findings in the translational setting is also advisable.
Minor comments:
- please change "hamartia-like lesions" into hamatomas/hamartomatous lesions throughout the text.
- higher resolution is needed for figure 2a; slightly higher magnification is advisable for figure 6 b&c.
Author Response
Tuberous sclerosis is a very interesting field of research. The authors establish a very interesting model of TSC in mice and show the efficacy of 5-azacytidine in case of epigenetical silencing of the tubers gene. The paper is globally well written and well presented. Discussion is consistent with the results. Methods are clearly described. Globally, Bernardelli et al made an excellent job. However, the authors state that their mouse model is ideal to follow the progression of TSC. A short sentence regarding the follicular localization of cutaneous hamartomas and the frequent involvement of the lungs in disseminated forms of the disease is needed for non-experts in the field both in the abstract and in the introduction. Since the two parameters considered in vivo are hair growth and skin thickening, a mention of the clinical relevance of such findings in the translational setting is also advisable.
We thank the reviewer for the appreciation of our manuscript. We added localization’s details for cutaneous hamartomas and lung involvement (line 42-43; 45-48) and mention regarding the clinical relevance (line 218-220; line 248-249).
Minor comments:
- please change "hamartia-like lesions" into hamatomas/hamartomatous lesions throughout the text.
We made the change throughout the text.
- higher resolution is needed for figure 2a; slightly higher magnification is advisable for figure 6 b&c.
As requested, the figures 2a, 6b and c are shown with higher magnification.
Reviewer 3 Report
The manuscript entitled “Primary TSC2-/meth cells induce follicular neogenesis in an innovative TSC mouse model” by Clara Bernardelli et al focuses on the development of a mouse model for skin manifestations by subcutaneous injection of TSC2-/meth cells in nude mice. The suggested mouse model reproduces the genetic and epigenetic complexity underlying the onset and the progression of skin TSC-related manifestations.
The authors claim that approach of primary cell injection is a suitable tool for the study of cutaneous TSC lesions. The work is interesting and up-to-date.
The method used is adequately described and the results obtained are presented well enough. Besides, the conclusions of the authors are supported by the clear graphs and pictures.
The results obtained are significant for a better understanding of the pathogenesis in question and can be used for further research.
The following critical remarks can be made.
1. Picture 2 with representative haematoxylin-eosin (left) and HMB45 immunohistochemistry (right) of skin sections from control mice is not clear enough.
2. The results of the research are obtained with the use of only one patient. There is some doubt as to whether the suggested protocol is reproduced.
3. According to ATCC the primary cultures are cells freshly isolated from organ tissue and maintained for growth in vitro. How long and how math passages had been culturing the primary TSC2-/meth cells before been transplanted to mouses? It is necessary to represent this information.
Author Response
The manuscript entitled “Primary TSC2-/meth cells induce follicular neogenesis in an innovative TSC mouse model” by Clara Bernardelli et al focuses on the development of a mouse model for skin manifestations by subcutaneous injection of TSC2-/meth cells in nude mice. The suggested mouse model reproduces the genetic and epigenetic complexity underlying the onset and the progression of skin TSC-related manifestations.
The authors claim that approach of primary cell injection is a suitable tool for the study of cutaneous TSC lesions. The work is interesting and up-to-date.
The method used is adequately described and the results obtained are presented well enough. Besides, the conclusions of the authors are supported by the clear graphs and pictures.
The results obtained are significant for a better understanding of the pathogenesis in question and can be used for further research.
The following critical remarks can be made.
- Picture 2 with representative haematoxylin-eosin (left) and HMB45 immunohistochemistry (right) of skin sections from control mice is not clear enough.
We improved the Figure 2a as also requested by the Reviewer 2.
- The results of the research are obtained with the use of only one patient. There is some doubt as to whether the suggested protocol is reproduced.
We thank the Reviewer for the observation that points out the data reproducibility. By using LAM/TSC cells, isolated from the chylous thorax of a LAM/TSC patient and bearing a TSC2 mutation and an epigenetic modification (Lesma et al, J. Cell. Mol. Med. Vol 18, 2014), we performed experiments following the same protocol as the one described in this manuscript. We obtained similar results as now reported in line 250-253. However, since LAM/TSC cells express peculiar properties such as senescent features, we are still characterizing this model more in deep and we hope to be able to obtain defined and relevant data for a new manuscript.
- According to ATCC the primary cultures are cells freshly isolated from organ tissue and maintained for growth in vitro. How long and how math passages had been culturing the primary TSC2-/meth cells before been transplanted to mouses? It is necessary to represent this information.
The cell passage numbers for TSC2-/meth cells have been included. We used the cells between 3 up to 5 passages (line 278-280).
This manuscript is a resubmission of an earlier submission. The following is a list of the peer review reports and author responses from that submission.
Round 1
Reviewer 1 Report
Major issues
#1. In Introduction, “brain astrocytoma” sounds brain parenchymal glioma. Please say intraventricular subependymal giant cell astrocytoma.
#2. In addition to surgical and topical mTOR inhibitor, please add mTOR inhibitor intake.
#3. Please consider using the word “hamartia” differentiating from “hamartoma”. The authors only used hamartoma that means a benign tumor . However, cutaneous lesions, brain cortical/subcortical tubers, LAM are not always tumor-like.
Minor issues
#1. Figure legends need consistency. In the text, the authors used capital, but in the legends lower-case.
Author Response
Major issues
#1. In Introduction, “brain astrocytoma” sounds brain parenchymal glioma. Please say intraventricular subependymal giant cell astrocytoma.
- We thank the reviewer for the punctual observation. We modified the text (line 40).
#2. In addition to surgical and topical mTOR inhibitor, please add mTOR inhibitor intake.
- It was added (line 50).
#3. Please consider using the word “hamartia” differentiating from “hamartoma”. The authors only used hamartoma that means a benign tumor. However, cutaneous lesions, brain cortical/subcortical tubers, LAM are not always tumor-like.
- Thank you for this observation We changed the term through the text.
Minor issues
#1. Figure legends need consistency. In the text, the authors used capital, but in the legends lower-case.
- We apologize for the discrepancy. The corrections have been made.
Reviewer 2 Report
This is an interesting manuscript that lacks substantial detailed results to back up the feeble results presented. It is worth pursuing further analysis of these data and the presentation of robust statistics for its preparation for publication. The authors did a good job in the investigational studies including design and performance but are short of analysis. However, the take-home message based on the result presented dampens enthusiasm as all figures provide mostly suboptimal data including figures and composites images that are not substantiated with alternative approaches as well as quantitation and statistical data analysis. I authors are therefore asked to add detailed data such as western blot for results presented in Figure 1. figures 2 -4 need western blots, quantification, and or immunoflurescence images. It is recommended to showblot low and high power mag zoomed images of the localization of the expression of the markers shown.
Author Response
This is an interesting manuscript that lacks substantial detailed results to back up the feeble results presented. It is worth pursuing further analysis of these data and the presentation of robust statistics for its preparation for publication. The authors did a good job in the investigational studies including design and performance but are short of analysis. However, the take-home message based on the result presented dampens enthusiasm as all figures provide mostly suboptimal data including figures and composites images that are not substantiated with alternative approaches as well as quantitation and statistical data analysis. I authors are therefore asked to add detailed data such as western blot for results presented in Figure 1. figures 2 -4 need western blots, quantification, and or immunofluorescence images. It is recommended to showblot low and high power mag zoomed images of the localization of the expression of the markers shown.
- We thank the Reviewer for the comments on our manuscript. Considering the observations and the opportunity to improve our communication, we analyzed and quantified the data reported in Figure 1, 3 and 4. For Figure 1 the PKH26 labelled cells have been counted and represented as the average number of PKH26-labelled cells in both mm2 and mm3 skin sections by considering the diameter of the cells and the section thickness.
- In view of the fact that in our sections only TSC2-/meth cells can show positivity to the human HMB45 antibody, the analysis of HMB45-positive cells to quantify the presence of the human injected TSC2-/meth cells would give the same information provided by the previous evaluation of PKH26 labelled cells. To demonstrate the specificity of HMB45 antibody for human cells, in Figure 2 we added the immunohistochemistry analysis with HMB45 antibody performed in skin sections of control mice. Moreover, the western blot analysis cannot be performed since the HMB45 antibody is not suitable for this technique. Compared to PKH26-labelling experiment, the HMB45-positivity allow us to evaluate the localization of TSC2-/meth cells that appears to be near the bulbs and alongside the mature hair.
- We quantified the phospho-S6 and phospho-Erk positive cells in the skin that are expressed as percentage by counting the positive cells on total number of cells. We hope that data reported in Figure 3 and Figure 4 now result more detailed and stronger considering also the performed statistical analysis (shown in the Supplemental Figure 2 and 3). We understand the suggestion to study the phosphorylation of S6 and Erk by western blot, however we think that the main result is the localization of the cells positive to phospho-S6 and phospho-Erk near hair bulbs and alongside mature hair and the uneven localization makes hard to obtain reliable results by western blot.
- To answer to the recommendation of the reviewer, we added low and high magnification of the images shown in Figure 2, 3 and 4 that are shown in the Supplemental figure 1, 2, and 3, respectively.
Round 2
Reviewer 2 Report
The authors did not address the flaws in the manuscript but provided mostly rebuttal which has not changed the current status. This could be an outright rejection if this manuscript does not receive significant revision.